# Musculoskeletal Changes in Hemophilia Patients Subsequent to COVID−19 Lockdown

**DOI:** 10.3390/healthcare9060702

**Published:** 2021-06-09

**Authors:** Rubén Cuesta-Barriuso, Javier Meroño-Gallut, Raúl Pérez-Llanes, Roberto Ucero-Lozano

**Affiliations:** 1Department of Physiotherapy, University of Murcia, 30100 Murcia, Spain; 2Royal Victoria Eugenia Foundation, 28029 Madrid, Spain; 3Tú. Bienestar 360°, Physiotherapy and Medical Center, 30730 San Javier-Murcia, Spain; ajmerono@gmail.com; 4Department of Physiotherapy, Catholic University San Antonio-UCAM, 30107 Murcia, Spain; rperez@ucam.edu; 5Department of Physiotherapy, European University of Madrid, 28670 Madrid, Spain; roberto.ucero@universidadeuropea.es

**Keywords:** hemophilia, COVID−19, joint disease, musculoskeletal pain, physiotherapy modalities

## Abstract

(1) Background. The lockdown period due to the COVID−19 pandemic has drastically decreased levels of physical activity in the population. Hemophilia is characterized by hemarthrosis that leads to chronic, progressive and degenerative joint deterioration. (2) Methods. This observational study recruited 27 patients with hemophilia and arthropathy. Knee, ankle and elbow joints were assessed. The frequency of clinical hemarthrosis, pain intensity, pressure pain threshold, and joint ROM were evaluated. (3) Results. Following lockdown, a significant deterioration of joint condition, perceived joint pain and range of motion was noted in all joints. There were no changes in the frequency of knee hemarthrosis, while the frequency of ankle hemarthrosis significantly reduced. However, the frequency of elbow hemarthrosis increased. Depending on the degree of hemophilia severity, there were changes in pressure pain threshold in the elbow and in pain intensity and range of motion of the ankle joint. According to the type of treatment, i.e., prophylaxis vs. on-demand treatment, there were differences in the joint condition in elbows and the plantar flexion movement of the ankle. There were no differences in the knee joint based on the severity of the disease, the type of treatment or the development of inhibitors (4). Conclusions. Because of the COVID−19 lockdown, the musculoskeletal status of patients with hemophilia deteriorated. Joint condition, perceived pain, and range of motion were significantly affected. The frequency of clinical hemarthrosis did not increase during this period. A more active therapeutic model could prevent rapid deterioration in patients with hemophilic arthropathy during prolonged sedentary periods.

## 1. Introduction

Severe Acute Respiratory Syndrome coronavirus 2 (SARS-CoV−2) emerged in the last quarter of 2019, leading to COVID−19 disease [1]. It quickly spread to other countries becoming a global pandemic [2]. The virus spreads easily among people through different forms of transmission such as close physical contact [3]. COVID−19 is a systemic disease capable of spreading throughout the bloodstream, affecting several organs and systems.

The World Health Organization declared the global pandemic due to COVID−19 in mid-March 2020. The Spanish Government declared a state of alarm, confining the entire population from March 14 to June 21, 2020. As a result of this lockdown, physical activity reduced significantly in the population [4]. National health systems have been forced to divert all their health care resources to the acute care of patients infected with this disease [5].

Hemophilia is characterized by the absence or deficiency of blood clotting factors. Depending on the missing factor, there are two types of hemophilia: hemophilia A (deficiency of clotting factor VIII) and hemophilia B (deficiency of clotting factor IX). According to the clotting factor levels in functional plasma, there are three categories of hemophilia severity: mild (5%–40%), moderate (1%–5%) and severe (<1%) [6].

There are two types of treatment strategies: on-demand treatment and prophylactic treatment, considered the gold standard for reducing bleeding and preventing effects in the joints [7]. Inhibitors are the main treatment complication in patients with hemophilia. These inhibitors disable the action of clotting factor concentrates rendering their administration useless.

Hemarthroses are hemorrhagic episodes that appear in the intra-articular space due to direct or indirect trauma. Hemarthrosis is the main clinical manifestation of hemophilia, appearing mainly in knees, elbows and ankles. The recurrence of hemarthrosis in the same joint causes progressive joint deterioration, leading to cartilage, bone and synovial membrane alterations. These intraarticular alterations together with the development of new hemarthroses generate synovial hypertrophy (hemophilic synovitis) leading to a degenerative and chronic lesion (hemophilic arthropathy). The main consequences of arthropathy include chronic pain, loss of joint functionality and limitations in the range of motion. These changes cause biomechanical and gait alterations, proprioception alterations and atrophy of the periarticular muscles. Without prophylactic treatment, joint damage develops at an early age in 85% of patients with severe hemophilia [8].

Joint health prior to onset, a low body mass index, limitation in the development of activities, the type of joint injury, the frequency of bleeding events and the development of synovitis are factors that are associated with the degree of joint deterioration [9]. However, Tagliaferri et al. [10] reported significant joint improvement in adult patients on prophylactic treatment compared to those receiving on-demand treatment, as well as a delay in the progression of radiological changes, measured on the Pettersson [11] scale, including those patients with significant joint damage.

The objective was to evaluate the frequency of clinical hemarthrosis, joint condition, perceived joint pain and range of motion in patients with hemophilic arthropathy following the COVID−19 lockdown period.

## 2. Materials and Methods

### 2.1. Study Design

This was a prospective observational study in patients with hemophilic arthropathy. This study was conducted between February and July 2020. The Research Ethics Committee of the Virgen de la Arrixaca University Hospital approved the project subject (id. 2020−2−9-HCUVA). The study was likewise registered with an international register (NCT04491318).

### 2.2. Participants

Patients were recruited from the physiotherapy clinic of the Spanish Federation of Hemophilia (CEE Fishemo). The study included patients diagnosed with hemophilia A and B; over 18 years of age; able to walk (with or without technical aids); and able to perform daily life activities without the help of others. Exclusion criteria were patients with neurological or cognitive disorders; patients who had been infected with COVID−19; and patients enrolled in a clinical trial with FVIII/FIX concentrates.

### 2.3. Outcome Variables

Evaluation at baseline (T0) was carried out in February 2020 and the post-treatment evaluation (T1) was carried out in July 2020. Four dependent variables were evaluated: frequency of clinical hemarthrosis, joint condition, perceived joint pain and range of motion. The same physiotherapist, blinded to study objectives, following the same protocol and the same measuring instructions, performed all assessments. Prior to the study, an interobserver reliability analysis was performed between the rater and one of the study researchers.

The frequency of bleeding was controlled by an online self-reporting system made available to patients at baseline. This report included a monthly schedule where patients recorded the date, the main symptoms (hotness, swelling, functional deficit, pain) and the joint affected by hemarthrosis [12]. Adherence to prophylactic treatment was measured with a self-report provided to the patients at the beginning of the study period. Using this record, the patients reported the days of clotting factor concentrate infusion and the basis for such infusion (prophylaxis or bleeding).

Hemophilia Joint Health Score (HJHS), version 2.1., was used to assess joint status [13]. This instrument assesses eight items: swelling (0–3); duration of swelling (0–1); muscle atrophy (0–2); joint crepitus (0–2); loss of flexion (0–3) and extension (0–3); joint pain (0–2) and muscle strength (0–4). In addition, there is an overall gait score (0–4). The maximum score among the six joints evaluated (knees, ankles and elbows) is 124 points (a higher score denotes a poorer joint condition). The score range for each joint is 0–20 points.

Perceived pain was measured with two measuring instruments: the visual analog scale and a pressure algometer (Wagner Force One ™ Digital Force Gage FDIX). The visual analog scale scores perceived pain ranging from 0 to 10 points (0 being no pain, and 10 being the maximum perceived pain). The pressure algometer bilaterally measures the pressure pain threshold at which the patient perceives pain at pressure at a given location when increasing this pressure at a speed of approximately 50 kPa/s. The unit of measurement is Newton/cm^2^. The pressure pain threshold was measured on the lateral epicondyle [14], 3 cm from the midpoint of the inner edge of the patella [15], and from the ventral region to the lateral malleolus [16].

The ranges of motion of the joints were measured with a goniometer. This measuring instrument has demonstrated good intra-and inter-observer reliability in the evaluated joints. Mobility in the elbow [17], knee and ankle [18] joints was evaluated.

### 2.4. Statistical Analysis

The statistical analysis was performed using SPSS software, version 19.0, for Windows (IBM Company, Armonk, NY, USA). The main descriptive statistics (median and interquartile range) were calculated. Sample distribution was analyzed using the Shapiro-Wilk test. The difference in means between the two assessments were calculated using the Paired Samples *t*-Test. The student’s *t*-test for independent samples was used to obtain the changes based on the type of treatment (prophylaxis or on demand) and the presence of antibodies (inhibitors). With a one-factor ANOVA test, changes were calculated depending on the degree of severity of hemophilia patients. The significance level of the study was α < 0.05.

## 3. Results

### 3.1. Participants

The median age of the 27 patients included in the study was 42 (IQR: 15) years, with a median body mass index of 26.42 (IQR: 4.59) kg/m^2^. There were changes in body weight between the two evaluations (T0 = 80.50 (16.00); T1 = 82.50 (14.25); *p* < 0.001). A total of 82.5% of patients were diagnosed with hemophilia A, the severe phenotype being the most common (70.4%). Only 25.9% of patients had inhibitors, while 70.4% of subjects received prophylactic treatment. Table 1 shows the descriptive characteristics of the patients.

### 3.2. Joint Changes

Following the confinement period, changes were observed in the joint state of elbows (*p* < 0.001), knees (*p* = 0.001) and ankles (*p* = 0.03). Range of motion decreased (*p* < 0.001) in all joints evaluated. Likewise, we noted a deterioration of the joint health in elbows (*p* < 0.01), knees (*p* < 0.01), and ankles (*p* = 0.03). In terms of pain, pain intensity increased (*p* < 0.001) in all joints and the pressure pain threshold dropped in elbows (*p* = 0.001), knees (*p* = 0.03) and ankles (*p* = 0.002). An increase in the frequency of clinical hemarthrosis was observed in the elbows (*p* = 0.04). However, the incidence of bleeding in the ankle joint decreased (*p* = 0.01). No changes were reported in the knee joint (*p* = 1.00). Table 2 shows the statistics for the joints evaluated.

When analyzing changes according to the degree of severity of hemophilia, differences were noted (*p* < 0.05) in pain intensity in the elbow (*p* = 0.001) and ankle (*p* = 0.009), and in dorsal (*p* = 0.02) and plantar (*p* < 0.001) flexion movements of the ankle. When the analysis was performed according to the type of treatment (on demand vs prophylactic), we found differences in the joint state of the elbow (*t* = 2.23; *p* = 0.03) and ankle plantar flexion (*t* = 2.36; *p* = 0.02). With regard to the analysis based on the development of inhibitors, there were differences in the frequency of ankle hemarthrosis (*t* = 2.16; *p* = 0.04). Table 3 shows the analyses of interaction according to hemophilia severity, the type of treatment, and the development of inhibitors.

## 4. Discussion

The aim of this study was to assess musculoskeletal complications derived from the COVID−19 stay-at-home period in adult patients with hemophilia and arthropathy. An overall deterioration was observed for most variables, mainly in range of motion and pain intensity. As a result of the onset of the COVID−19 pandemic, the scientific community has significantly developed and increased the number of studies that evaluate and quantify the impact of the disease [19].

The prophylactic treatment adherence rate during confinement was 86%. Following the lockdown period, no changes were reported in the frequency of hemarthrosis in knees, while the number of bleeding events in the ankle joint reduced. This reduced frequency of ankle joint bleeding could be related to adherence to treatment and the reduction of possible injury events (trauma, increased load or excessive walking, etc.) due to the drastic reduction of activity during the stay-at-home order [4]. Stephensen et al. [20] reported a high frequency of hemarthrosis in those patients who play sports, having a greater impact on the ankle. This activity increases the frequency of hemarthrosis of the ankle, as compared to knees and elbows. The patients in our study, subjected to a prolonged period of sedentary lifestyle, experienced a decrease in the frequency of hemarthrosis in the ankle joint. Increased frequency of elbow hemarthrosis could be due to the higher levels of manual and recreational activity, as well as everyday household activities, compared to decreased lower limb activity.

Joint health of patients with hemophilic arthropathy deteriorated during lockdown in all three joints. Under normal conditions, changes in the joint health of patients with hemophilic arthropathy do not take place in such a short period of time. A number of studies with follow-up periods of 5 to 10 years reported minimal changes in joint health in young adults with hemophilia, measured on the HJHS9, the Orthopedic joint score of the World Federation of Hemophilia [21] and the Pettersson Radiological score [22].

Recent studies have reported no changes in the joint condition of elbows [23], knees [24] or ankles [25] in patients in the control group who, without any intervention, continued with their usual daily activities over a 15 to 24-week period. This deterioration could be related to the decrease in range of motion that causes intraarticular changes and alterations in neighboring tissues [26]. Similarly, these changes may also be due to significantly increased weight of patients during this period. The increase in weight and the deteriorated joint condition have already been described by Chang et al. [27], who noted a correlation between overweight and joint health indices.

Patients have experienced a worsening in pain-related variables. Pain is described as a sensitive and emotional experience that patients perceive as slightly or very unpleasant, associated with real or potential tissue damage, or described in terms of the injury in itself [28]. This sensory and emotional experience has an anticipatory and evaluative component [29] influenced by the information received by the brain from tissues and the environment. This information is evaluated in relation to our previous experiences [30]. In the context of a pandemic, much of the information being received is about danger. If this information is further accompanied by the confrontation with an unknown situation and isolation, an increase in anxiety and depressive symptoms is generated [31], causing a stimulation of the prevention response [32]. All these events may increase the perception of pain. On the other hand, the hypervigilance generated by anxiety [32] itself and the central and peripheral sensitization associated with pain [33] might explain the lowering of pressure pain thresholds in the different joints.

This lockdown situation has drastically reduced social relationships and activities, preventing any positive effect on patients [34]. Alleviating affective symptoms is essential for patients with chronic pain [35]. Therefore, the increased perception of pain noted in this study might be related to the absence of affective symptoms derived from the long stay-at-home term due to the COVID−19 pandemic.

Inactivity promotes decreased range of motion in the especially affected diarthrodial joints in patients with hemophilia. Likewise, the involvement of periarticular structures such as capsule, tendons, and ligaments, can induce movement alterations [26]. At the muscular level, decreased activity also induces changes in muscle structure and function. At the structural level, muscle atrophy tends to appear, especially affecting antigravity muscles, mainly Type I fibers. In addition, there is a decrease in the number of Type I fibers, increasing the number of Type II fibers. Atrophy leads to decreased protein synthesis and increased degradation thereof. At the functional level, maximum strength and maximum power are reduced, although the contraction speed increases to compensate for this loss of strength [36].

The various inactivity-related factors (increased pain and weight, and tissue and functional changes in the musculoskeletal system) can cause changes in the movement pattern of these patients [30], modifying their motor recruitment [36] patterns and quality of movement.

### 4.1. Limitations

The main limitation of this study is the failure to measure psychosocial variables such as perceived quality of life, catastrophism, anxiety or depression. Their evaluation could widen our understanding of the situation of these patients in a context of confinement. Another limitation is the failure to assess joint functionality, something that would allow us to evaluate joint deterioration in these patients. Finally, only patients from Spain have been included, which may make it difficult to extrapolate the results to other countries subjected to varying degrees of confinement.

### 4.2. Perspectives

Public health services should promote research to establish the relationship between the clinical situation of patients with hemophilia and confinement to homes during the quarantine periods taking place worldwide. Similarly, factors which may mitigate these adverse effects must be identified. Thus, it is important to develop aspects such as: (i) the optimal way to carry out physical activity during lockdown; (ii) the best tools to improve the quality of life of patients after confinement; and (iii) preventive measures against a new wave of COVID−19 or other potential pandemics [37].

The total interruption of everyday activities in this population of patients adversely affects pain and mobility. The prolonged period of home confinement due to the COVID−19 pandemic made face-to-face intervention models impossible. This should prompt us to address the importance of introducing remote physiotherapy measures in order to maintain a minimum control on the physical activity of our patients and the course of their degenerative joint processes. Access to work models that allow us to offer feedback on their implementation would strengthen the model for the involvement of patients as active subjects taking part in their treatment process, educating them and promoting their commitment.

## 5. Conclusions

Lockdown due to the COVID−19 pandemic deteriorated joint health, increased pain intensity, and reduced pressure pain threshold and range of motion in adult patients with hemophilic arthropathy. The frequency of hemarthrosis reduced in cases of ankle arthropathy, while increasing in the elbow joint after the three-month lockdown period.

The treatment model for these patients needs to be changed, reinforcing their early and active treatment. The promotion of educational programs and access to non-face-to-face interventions in situations in which on-site assistance is not possible could help to prevent the rapid deterioration of joint health in patients with hemophilic arthropathy.

## Figures and Tables

**Table 1 healthcare-09-00702-t001:** Median (and interquartile range) of the descriptive characteristics of the patients with hemophilia included in the study.

	Variables	Mean (SD)
Anthropometric variables	Age (year)	37.42 (2.27)
	Height (cm)	173.37 (1.29)
	Weight before lockdown (Kg)	80.50 (16.00)
	Weight after lockdown (Kg)	82.50 (14.25)
	Treatment adherence (%)	87.37 (1.29)
		***n* (%)**
Clinical variables	Type of hemophilia (A/B)	23/4 (85.2/14.8)
	Severity of hemophilia (Mild/Moderate/Severe)	1/7/19 (3.7/25.9/70.4)
	Treatment (On demand/Prophylactic)	8/19 (29.6/70.4)
	Development of inhibitors (No/Yes)	20/7 (74.1/25.9)

**Table 2 healthcare-09-00702-t002:** Descriptive statistics (mean and standard deviation), confidence interval and changes in all joints after the quarantine period.

Joint	Variables	T0	T1	95% CI	Sig.
Elbow	Hemarthrosis (number)	0.19 (0.51)	0.33 (0.61)	−0.29; −0.00	0.04 *
	Joint status (range 0–20)	6.30 (4.35)	6.72 (4.11)	−0.62; −0.23	0.00 **
	Pain intensity (range 0–10)	1.11 (2.04)	1.63 (2.09)	−0.72; −0.30	0.00 **
	Pressure pain threshold (N)	70.75 (21.09)	70.07 (21.74)	0.27; 1.06	0.00 *
	Flexion (degrees)	130.43 (9.73)	129.33 (10.01)	0.59; 1.59	0.00 **
	Loss of extension (degrees)	16.83 (17.39)	18.00 (17.50)	−1.55; −0.78	0.00 **
Knee	Hemarthrosis (number)	0.35 (0.73)	0.35 (0.75)	−0.13; 0.13	1.00
	Joint status (range 0–20)	4.67 (4.80)	4.94 (5.02)	−0.43; −0.12	0.00 *
	Pain intensity (range 0–10)	1.17 (1.34)	1.67 (1.93)	−0.76; −0.23	0.00 **
	Pressure pain threshold (N)	63.63 (19.31)	63.31 (19.46)	0.02; 0.60	0.03 *
	Flexion (degrees)	131.59 (4.37)	130.70 (5.23)	0.54; 1.23	0.00 **
	Loss of extension (degrees)	2.22 (2.66)	3.06 (3.55)	−1.17; −0.49	0.00 **
Ankle	Hemarthrosis (number)	0.24 (0.51)	0.11 (0.31)	0.02; 0.23	0.01 *
	Joint status (range 0–20)	7.94 (3.94)	8.15 (3.94)	−0.39; −0.01	0.03 *
	Pain intensity (range 0–10)	1.67 (1.41)	3.26 (2.45)	−1.98; −1.19	0.00 **
	Pressure pain threshold (N)	62.06 (15.65)	61.68 (15.89)	0.10; 0.66	0.00 *
	Dorsal flexion (degrees)	5.00 (4.98)	3.89 (5.51)	0.68; 1.53	0.00 **
	Plantar flexion (degrees)	25.69 (10.45)	24.44 (10.98)	0.85; 1.63	0.00 **

T0: baseline assessment; T1: posttreatment assessment; 95% CI: 95% confidence interval; Sig.: significance. * Significant difference (*p* < 0.05). ** Significant difference (*p* < 0.001).

**Table 3 healthcare-09-00702-t003:** Changes according to the degree of hemophilia severity, type of treatment and the development of inhibitors.

Joint	Variable	Severity of Hemophilia		Type of Treatment		Development of Inhibitors	
		F (Sig.)	ANOVA (Sig.)	F (Sig.)	t (Sig.)	F (Sig.)	t (Sig.)
	Hemarthrosis	8.61 (0.00) †	2.25 (0.23)	0.06 (0.79)	−0.76 (0.44)	2.78 (0.10)	−0.54 (0.59)
	Joint status	1.80 (0.17)	1.78 (0.17)	1.17 (0.28)	2.23 (0.03) *	0.58 (0.44)	1.75 (0.08)
Elbow	Joint pain intensity	3.17 (0.05)	0.63 (0.53)	1.65 (0.20)	1.44 (0.15)	0.45 (0.50)	−1.10 (0.27)
	Pressure pain threshold	2.08 (0.13)	8.20 (0.00) *	5.64 (0.02) ‖	−0.58 (0.56)	0.09 (0.92)	0.33 (0.74)
	Flexion	0.64 (0.53)	1.86 (0.16)	0.16 (0.69)	−1.22 (0.22)	0.55 (0.46)	0.11 (0.90)
	Loss of extension	2.28 (0.11)	0.70 (0.50)	0.23 (0.62)	0.07 (0.94)	0.04 (0.83)	−0.58 (0.56)
	Hemarthrosis	2.61 (0.08)	1.17 (0.31)	1.25 (0.26)	−0.62 (0.53)	10.18 (0.00) ‖	−0.91 (0.37)
	Joint status	1.61 (0.20)	0.63 (0.53)	0.11 (0.74)	−0.23 (0.81)	11.97 (0.00) ‖	−1.85 (0.08)
Knee	Joint pain intensity	42.69 (0.00) †	2.83 (0.06)	0.04 (0.83)	−0.92 (0.36)	0.47 (0.49)	0.63 (0.52)
	Pressure pain threshold	0.34 (0.71)	2.41 (0.10)	2.33 (0.13)	0.33 (0.74)	0.08 (0.77)	−0.96 (0.33)
	Flexion	0.29 (0.74)	1.06 (0.35)	0.64 (0.42)	−0.41 (0.67)	7.42 (0.00) ‖	1.96 (0.06)
	Loss of extension	12.32 (0.00) †	1.98 (0.32)	2.43 (0.12)	−0.55 (0.58)	0.57 (0.45)	−0.33 (0.74)
	Hemarthrosis	10.49 (0.00) †	1.26 (0.29)	0.13 (0.71)	0.81 (0.41)	16.14 (0.00) ‖	2.16 (0.04) *
	Joint status	1.83 (0.17)	0.95 (0.39)	2.62 (0.11)	−1.90 (0.06)	5.06 (0.02) ‖	1.03 (0.30)
Ankle	Joint pain intensity	0.44 (0.64)	5.21 (0.00) *	0.28 (0.59)	−1.13 (0.26)	0.80 (0.37)	0.27 (0.78)
	Pressure pain threshold	1.10 (0.33)	0.83 (0.43)	0.42 (0.51)	−0.32 (0.74)	0.15 (0.70)	−0.75 (0.45)
	Dorsal flexion	1.00 (0.37)	3.86 (0.02) *	0.52 (0.47)	0.71 (0.47)	0.79 (0.37)	0.48 (0.63)
	Plantar flexion	3.23 (0.04) †	9.84 (0.00) **	1.53 (0.22)	2.36 (0.02) *	2.60 (0.11)	0.13 (0.89)

t: Levene statistics; ANOVA: Analysis of variance statistics; †: Brown-Forsythe test; ‖: Mann-Witney U test. Sig.: significance. * Significant difference (*p* < 0.05). ** Significant difference (*p* < 0.001).

## Data Availability

The data presented in this study are available on request from the corresponding author. The data are not publicly available due to privacy.

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
