# Peer review of "Musculoskeletal Changes in Hemophilia Patients Subsequent to COVID−19 Lockdown"

_healthcare, 2021, doi:10.3390/healthcare9060702_

Round 1

Reviewer 1 Report

The paper by Cuesta-Barriuso et al. discusses the important issue of the impact of COVID-19 lockdown on the musculoskeletal status of hemophilia patients. Overall, the paper successfully demonstrates the negative impact of a less active lifestyle imposed by the pandemic lockdown through observing 27 hemophilia patients. There is room for improvement in the phrasing and clarity of some sections of the paper, some concerns are highlighted below in the order they appear:

  1. In the abstract, please mention the six joints being evaluated during the study and rephrase the results section (lines 18 to 23) to highlight the state of each joint being monitored. For example, in line 19-20 the sentence starting with “Only…” is confusing, it states that the frequency of hemarthrosis in the ankles improved although this is opposite to what is stated in table 2. Please clarify and rephrase as needed. The rest of the section after this sentence discusses the elbow and ankle joints and omits mentioning the knees, please consider clarifying and rephrasing. (see comment 11 and 12)

  2. The abstract conclusion (lines 27-28) could be rephrased to deliver the message in a clearer way without commenting on the ‘feasibility’ of the suggested solution as different settings with different circumstances might not all perceive feasibility similarly.

  3. Line 35 “while interpersonal proximity is an important factor”, I suggest rephrasing to clarify, while in the middle of the sentence could be confusing.

  4. Lines 38-39 the verb tense (declares/declared) changes between present and past, please revise the grammar and ensure consistency throughout the paper.

  5. Line 48, if the three categories are all referring to both types of hemophilia, please remove FVIII from the mild category definition.

  6. Line 49, please rephrase to “There are two types of treatment approaches/strategies”.

  7. Line 54, please start the sentence with defining hemarthrosis for readers who might not be familiar with the term.

  8. Line 65, please consider rephrasing to signpost the sentence before enumerating the different factors.

  9. Lines 72-74, please consider rephrasing to “ The objective of this study was to evaluate the impact of the COVID-19 lockdown on the frequency of clinical hemarthrosis, pain intensity..etc”. On a different note, these objectives should be consistent with the outcome variables discussed in lines 90-91.

  10. Table 1, please highlight in the methods section how the treatment adherence was measured during the study.

  11. Lines 144-145, the sentence starting with “Finally the frequency of hemarthrosis in elbows increased, improving the incidence of bleeding in the ankle” is very confusing and not clear how the first part of the sentence relates to the second, please revise and rephrase.

  12. Table 2 shows significance of 0.01 for hemarthrosis (number), yet in the abstract, results and discussion the authors state that the number of bleeding events in the ankle improved. Please clarify.

  13. Lines 178-180, the relation between the findings of this study and those of Stephensen’s are not clear, please clarify why your findings could be similar to those of a study participating in sports and activities?

  14. Line 194, table 2 shows an average increase in weight of 1.26 kg, yet in line 194 the authors explain the deterioration of joint status by significant increase in weight, which is not the case in this study, please correct or justify accordingly.

  15. Line 201, typo received by the brain instead of receives.

  16. Lines 210-211, please consider rephrasing to a more scientific language, especially the term “happiness”.

  17. Line 226, reference 36 is missing the brackets.

  18. Line 254, please consider rephrasing to “COVID-19 pandemic led to a deterioration…”.

  19. References from 38 to 47 are not understandable.

Author Response

Reviewer 1

The paper by Cuesta-Barriuso et al. discusses the important issue of the impact of COVID-19 lockdown on the musculoskeletal status of hemophilia patients. Overall, the paper successfully demonstrates the negative impact of a less active lifestyle imposed by the pandemic lockdown through observing 27 hemophilia patients. There is room for improvement in the phrasing and clarity of some sections of the paper, some concerns are highlighted below in the order they appear:

  • In the abstract, please mention the six joints being evaluated during the study and rephrase the results section (lines 18 to 23) to highlight the state of each joint being monitored. For example, in line 19-20 the sentence starting with “Only…” is confusing, it states that the frequency of hemarthrosis in the ankles improved although this is opposite to what is stated in table 2. Please clarify and rephrase as needed. The rest of the section after this sentence discusses the elbow and ankle joints and omits mentioning the knees, please consider clarifying and rephrasing. (see comment 11 and 12). We have made the changes indicated by the reviewer in order to facilitate their understanding, providing the required information and rewriting the most difficult aspects of understanding.
  • The abstract conclusion (lines 27-28) could be rephrased to deliver the message in a clearer way without commenting on the ‘feasibility’ of the suggested solution as different settings with different circumstances might not all perceive feasibility similarly. As the reviewer points out, we have modified the conclusions to be more specific.
  • Line 35 “while interpersonal proximity is an important factor”, I suggest rephrasing to clarify, while in the middle of the sentence could be confusing. To avoid confusion for the reader, we have modified the text to facilitate understanding.
  • Lines 38-39 the verb tense (declares/declared) changes between present and past, please revise the grammar and ensure consistency throughout the paper. A complete grammar check of the text has been done by a professional translator.
  • Line 48, if the three categories are all referring to both types of hemophilia, please remove FVIII from the mild category definition. As the reviewer points out, all three categories refer to both types of hemophilia. We have removed "FVIII" from the definition of mild category.
  • Line 49, please rephrase to “There are two types of treatment approaches/strategies”. The term "treatment strategies" has been incorporated, which is the most appropriate to describe the two types of pharmacological treatments in patients with hemophilia.
  • Line 54, please start the sentence with defining hemarthrosis for readers who might not be familiar with the term. On the recommendation of the reviewer, we have included a brief description of hemarthrosis at the beginning of the paragraph.
  • Line 65, please consider rephrasing to signpost the sentence before enumerating the different factors. As the reviewer indicates, the phrase has been reformulated to facilitate understanding.
  • Lines 72-74, please consider rephrasing to “ The objective of this study was to evaluate the impact of the COVID-19 lockdown on the frequency of clinical hemarthrosis, pain intensity..etc”. On a different note, these objectives should be consistent with the outcome variables discussed in lines 90-91. As the reviewer points out, the study objectives have been rewritten, according to the nomenclatures indicated in the Outcome variables
  • Table 1, please highlight in the methods section how the treatment adherence was measured during the study. As indicated by the reviewer, we have included in the Methods section how adherence to treatment was measured.
  • Lines 144-145, the sentence starting with “Finally the frequency of hemarthrosis in elbows increased, improving the incidence of bleeding in the ankle” is very confusing and not clear how the first part of the sentence relates to the second, please revise and rephrase. We have rewritten the sentence to make the text easier to understand. According to what was indicated by the reviewer, we have separated the text into two independent sentences.
  • Table 2 shows significance of 0.01 for hemarthrosis (number), yet in the abstract, results and discussion the authors state that the number of bleeding events in the ankle improved. Please clarify. The significant result was a significant decrease in the number of hemarthros during the study period. For this reason, we speak of an improvement in the frequency of hemarthrosis in the ankle (in contrast to the worsening in the elbow joint, where the significant change is negative). For this reason, Table 2 was included, where the mean and standard deviation of all the variables are observed in the two evaluations carried out.
  • Lines 178-180, the relation between the findings of this study and those of Stephensen’s are not clear, please clarify why your findings could be similar to those of a study participating in sports and activities? According to the reviewer's annotation, this sentence could be misleading. To make it easier for the reader to understand, we have rewritten the text.
  • Line 194, table 2 shows an average increase in weight of 2.50 kg, yet in line 194 the authors explain the deterioration of joint status by significant increase in weight, which is not the case in this study, please correct or justify accordingly. In the text (line 136, in the Results section) the changes between both evaluations in weight are indicated, the difference being significant.
  • Line 201, typo received by the brain instead of receives. The type indicated by the reviewer has been corrected.
  • Lines 210-211, please consider rephrasing to a more scientific language, especially the term “happiness”. To avoid confusion for the reader, we have redrafted the text, as indicated by the reviewer.
  • Line 226, reference 36 is missing the brackets. The indicated reference has been corrected.
  • Line 254, please consider rephrasing to “COVID-19 pandemic led to a deterioration…”. To avoid confusion for the reader, we have redrafted the text, as indicated by the reviewer.
  • References from 38 to 47 are not understandable. As indicated by the reviewer, those references that do not correspond to the study, but to the example model of the Healthcare journal, have been eliminated.

Reviewer 2 Report

The article titled "Mucoskeletal changes in hemophilia patients subsequent to COVID-19 lockdown" by Cuesta-Barriuso et al., is a timely and important report. The authors have described how lockdown in Spain during COVID-19 affected patients with hemophilia and argue for for better approach for situations like these. 

The manuscript is well written. However, there are few minor concerns

1) It is not clear which dates are considered as T0 and T1?

2) The controls have not been explained clearly in this study? 

3) Minor English correction at few places is needed. 

Author Response

Reviewer 2

The article titled "Mucoskeletal changes in hemophilia patients subsequent to COVID-19 lockdown" by Cuesta-Barriuso et al., is a timely and important report. The authors have described how lockdown in Spain during COVID-19 affected patients with hemophilia and argue for for better approach for situations like these.

The manuscript is well written. However, there are few minor concerns

  • It is not clear which dates are considered as T0 and T1? As the reviewer indicates, at the beginning of the Outcome variables section the dates of both assessments (February and July 2020) have been exposed.
  • The controls have not been explained clearly in this study? In this study there is no control group. The comparison with control groups is made with respect to other studies, to observe the evolution of joint deterioration in patients with hemophilia under normal conditions, compared to the joint deterioration observed during confinement due to the COVID-19 pandemic.
  • Minor English correction at few places is needed. The text has been revised by a native English translator to correct the grammatical errors existing in the first edition of the text.